# Effect of Forest Users’ Stress on Perceived Restorativeness, Forest Recreation Motivation, and Mental Well-Being during COVID-19 Pandemic

**DOI:** 10.3390/ijerph19116675

**Published:** 2022-05-30

**Authors:** Don-Gak Lee, Jin-Gun Kim, Bum-Jin Park, Won Sop Shin

**Affiliations:** 1Graduated Department of Forest Therapy, Chungbuk National University, Cheongju 28644, Korea; don0810@naver.com; 2Korea Forest Therapy Forum Incorporated Association, Cheongju 28644, Korea; k64804171@gmail.com; 3Department of Environment & Forest Resources, Chungnam National University, Daejeon 34134, Korea; bjpark@cnu.ac.kr; 4Department of Forest Sciences, Chungbuk National University, Cheongju 28644, Korea

**Keywords:** stress, perceived restorativeness, forest recreation motivation, mental well-being, multi-group analysis, COVID-19 pandemic

## Abstract

Even though the COVID-19 pandemic has discouraged travel and people’s movements, the number of visitors to forests near cities which are easily accessible by private vehicle is increasing in Korea. This study aims to investigate the relationship between stress, perceived restorativeness, forest recreation motivation, and the mental well-being of forest users. A survey of forest users was conducted at three recreational forests near Seoul in the summer of 2020. A total of 1196 forest users (613 males and 583 females) participated in the study. As a result of the data analysis, it was found that stress had a negative correlation with perceived restorativeness, forest recreation motivation, and mental well-being; perceived restorativeness had a positive correlation with mental well-being, and forest recreation motivation had a positive correlation with mental well-being. For the relationship between stress and mental well-being, the fitness index that was mediated by the perceived restorativeness and the forest recreation motivation found that the model was statistically suitable. Through this study, a research model was derived that, if the stress of forest users is reduced, direct or indirect effects on perceived restorativeness, forest recreation motivation, and mental well-being are increased. Further, a multi-group analysis found that the effect of perceived restorativeness and forest recreation motivation on the mental well-being of the male group was higher than the effect on the female group. Using this research model to find ways to promote health in forests can be utilized for forest management or forest healing.

## 1. Introduction

Currently, 54% of the world’s population lives in urban areas. This figure has increased by more than 30% since 1950, and by 2050, 66% of the world’s population is expected to be urban [1]. From a psychological point of view, the urban lifestyle increases the demand for cognitive activities such as working and maintaining social relationships [2]. Humans use their resources and energy to relieve stressful situations; however, persistently relying on this method can cause disease. Low levels of stress can be tolerated. Still, when the severity of the stress increases or persists, it destroys biological systems. This is described as the “fight-or-flight response” [3]. Differences in individual responses to stressful situations are likely to lead to changes in immune function and the spread of disease [4,5] to support health and well-being [6]. In particular, forests have been reported to have physiological and psychological recovery effects; for example, by being in contact with a forest environment for just 15 min. Furthermore, due to the COVID-19 pandemic, the daily life of people around the world has changed, and it has had a significant impact on physical, mental, social, and economic well-being [7]. In particular, 47.5% of Koreans experience anxiety and depression [8] due to the continuation of the pandemic, and 14.9% of Seoul citizens are found to belong to the highly stressed group [9]. Of course, American citizens also experience stress in all its variations [7]. In other words, COVID-19 infection has become a multifaceted stressor, causing a physical, social, and economic loss for people worldwide [10,11]. According to the WHO [12], COVID-19 stress is caused by fear and worry, the way in which people respond to real threats, and discomfort with uncertainty or unknown situations. Therefore, stress management and healthy lifestyles are critical to public health.

Recently, making use of the natural environment has emerged as an external strategy to cope with stress. According to the Attention Restoration Theory, an intentional demand for attention is related to attention fatigue. Thus, spending time in a natural environment with many recovery factors can restore energy levels [13,14], and living with nature can promote recovery and healing. Utilizing the natural environment can also provide new opportunities to reduce elevated cortisol levels in stressful environments, and improve lowered immunity [15,16]. People under stress had lower stress levels in forest environments than in urban settings, and just strolling in the forest provided stress relief and greater mental stability [17,18]. In addition, it has been reported that enjoying activities in the forest positively affects feelings of stability, and is particularly effective in enhancing children’s sense of well-being [19,20].

Previous studies have reported that spending time in the forest is associated with various effects. For example, forest users experience reduced stress by escaping from daily life [21]. Forest environments increase perceived restorativeness, relieve stress, and induce a positive emotional effect [22,23]. In addition, exposure to forests is positively related to emotional and psychological well-being (PWB) [24,25], and it has been reported that the adverse effects of stress are alleviated when the ability to focus is restored by contact with the forest environment [23].

Spending time in the forest is also closely related to forest recreation motivation. The motivation for visiting the forest may be the expectation of improved mental and social health. We obtain these benefits through participation in forest activities, and the natural environment can contribute to our mental health and well-being [26,27]. Another important motivation for people to engage in activities in the natural environment might be to enjoy and be close to nature [28]; in other words, to enjoy the beautiful scenery and the scents and sounds of nature [29]. Moreover, the presence of hiking trails in certain parks, where users’ physical fitness motivations and natural area preferences may match, are associated with higher levels of physical activity [27].

However, despite these many previous studies, empirical studies on the relationship between stress and perceived restorativeness, forest recreation motivation, and the mental well-being of forest users are insufficient. Therefore, this study aims to investigate the relationship between stress and perceived restorativeness, forest recreation motivation, and the mental well-being of forest users. Specifically, the relationship between stress and the perception of the natural environment for recovery, forest recreation motivation, and mental well-being is investigated through structural equations, and the difference is identified through a multi-group analysis, according to the gender of forest users.

Therefore, we established a series of research hypothesis to perform for the purpose of this study. The research hypothesis is as follows:

**Hypothesis** **1** **(H1-1).**Stress will have a negative effect on perceived restorativeness.

**Hypothesis** **1** **(H1-2).**Stress will negatively affect the forest recreation motivation.

**Hypothesis** **1** **(H1-3).**Stress will negatively affect mental well-being.

**Hypothesis** **2** **(H2).**Perceived restorativeness will have a positive effect on mental well-being.

**Hypothesis** **3** **(H3).**The forest recreation motivation will have a positive effect on mental well-being.

**Hypothesis** **4** **(H4).**Perceived restorativeness will mediate the effect of stress on mental well-being

**Hypothesis** **5** **(H5).**Forest recreation motivation will mediate the effect of stress on mental well-being.

## 2. Methods

### 2.1. Study Subject and Research Method

This study was conducted by people who visited Mt. Gwanak Urban Forest, Mt. Bukhan National Park, and Mt. Yumyeong Natural Recreation Forest in the Seoul Metropolitan Area and the Gyeonggi-do province (Figure 1). Since it was impossible to control the movement of forest users, the survey was conducted for adults who were 19 years of age or older, wearing a mask, and descending from the midpoint of each forest on weekends and weekdays between 1 May and 15 July 2020, from 9 am to noon and from 2 to 5 pm. The average temperature of the site was 20–25 °C (Refer to an article [22] for detailed information related to the location of the study).

### 2.2. Measurements

#### 2.2.1. Psychosocial Well-Being Index-SF

The PWI-SF that was developed by Chang [30] is an 18-item Likert-4 scale used to measure the physical and psychological state experienced or felt by a person in recent weeks. The PWI-SF gives each item a score of (0~3) and the sum of the scores for each item is distributed between 0–54 points: up to 8 points (health group); <9–26 points; (potential stress group); <27–54 points (high-risk group). A higher score indicates a higher stress level [31]. As a result of analyzing the reliability of social and psychological stress, Cronbach’s α was 0.88.

#### 2.2.2. Perceived Restorativeness Scale

The PRS that was developed by Hartig et al. [32] is a tool to measure the psychological recovery effect of the recovery environment. This scale is a Likert-7 scale consisting of 4 factors, including 16 items: two items of ‘being away’; five items of ‘fascination’; four items of ‘coherence’; and five items of ‘compatibility’. A scale that was adapted by Kim [33] was used in this study. As a result of reliability analysis, Cronbach’s α was found to be: ‘being away 0.85’; ‘fascination 0.74’; ‘coherence 0.60’; and ‘compatibility 0.87’. The overall score was 0.87.

#### 2.2.3. Forest Recreation Motivation Scale

The FRMS that was developed by Yeon [34] is a tool to measure users’ motivation to manage recreation resources and their recreation experience. This scale is a Likert-5 scale consisting of 22 items and five factors: ‘nature-friendly motive’, six items; ‘social-friendly motive’, seven items; ‘daily life change motive’, four items; ‘self-exploration motive’, three items; and ‘natural achievement motive’, two items. As a result of analyzing the reliability with the FRMS, Cronbach’s α was ‘nature-friendly motive 0.75’; ‘social-friendly motive 0.86’; ‘motivation to pursue changes in daily life 0.65’; ‘self-exploration motive 0.62’; and ‘nature achievement motive 0.74’. The overall score was 0.91.

#### 2.2.4. Mental Well-Being Scale (K-MHC-SF: Korean Version of the Mental Health Continuum Short Form)

According to the happiness theory, the mental well-being scale that was developed by Keyes [35,36] is a tool to measure a person’s level of mental well-being. This scale is a Likert-6 scale consisting of 14 items and three factors: emotional well-being (questions 1–3); psychological well-being (questions 9–14); and social well-being (questions 4–8). As a result of analyzing trust using the mental well-being scale that was adapted by Lim et al. [37], Cronbach’s α was ‘emotional well-being 0.94’; ‘well-being 0.87’; and ‘Psychological well-being 0.93,’. The overall score was 0.95.

### 2.3. Research Model and Hypothesis

Structural equations are used to verify the stress, perceived restorativeness, motivation for forest recreation, and the mental well-being of forest users.

The research model is identified by setting the hypothesis as follows (Figure 2).


**Hypothesis 1.**
(H1-1).Stress will have a negative effect on perceived restorativeness. The natural environment is emerging as an external strategy for coping with stress. It is said that stress should be immediately alleviated and healed, and the forest is the best place for this [38]. Forest users experience stress reduction [21] and they can escape from daily life by enjoying a restorative environment [13]. Therefore, this study aims to verify the relationship between stress and the perceived restorativeness of forest users.(H1-2).Stress will negatively affect the forest recreation motivation. Outdoor recreation motivates people to escape from excessive work stress [39,40,41]. Leisure activities in the natural environment help people to cope with stress and are crucial for satisfying several needs. Therefore, in leisure activities in an outdoor setting, recovery from stress is the main recognized effect [42,43]. This study aims to verify the relationship between stress and the forest recreation motivation of forest users.(H1-3).Stress will negatively affect mental well-being. Stress is negatively correlated with well-being [44], and old-age stress threatens the happiness of the elderly [45]. Therefore, stress management helps people to cope with negative situations and positively affects their mental well-being [46]. Contact with the natural environment for 15 min promotes a reduction in cortisol concentration, which rises during periods of stress [16] and increases human immunity. A simple slow stroll helps to relieve stress and encourages mental stability [17], so we want to verify the relationship between stress and the mental well-being of forest users.
**Hypothesis 2.** Perceived restorativeness will have a positive effect on mental well-being. Exposure to nature positively correlates with emotional and psychological well-being (PWB) [25,47]. When the ability to pay attention is restored by contact with the natural environment it has a positive emotional effect by alleviating the adverse effects of stress. In particular, the effect of emotional enhancement varies depending on the level of exposure to the recovery environment effect [23]. This study aims to verify the relationship between forest users’ perceived restorativeness and mental well-being.**Hypothesis 3.** The forest recreation motivation will have a positive effect on mental well-being. An important motivation for people to engage in outdoor activities in their natural environment is to enjoy be close to nature [28]. The motivation of park visitors to restore their focus and experience comfort while spending time in nature encourages their interaction with nature [48]. Thus, the park environment can contribute to mental health and well-being by allowing visitors to participate in and benefit from recreational activities in all areas within the park [27]. This study aims to verify the relationship between forest recreation motivation and the mental well-being of forest users.**Hypothesis 4.** Perceived restorativeness will mediate the effect of stress on mental well-being. Nature has a stress-buffer effect on psychological well-being [49]. To solve stressful situations, people tend to accept their environment by facilitating positive perceptions about it or devising coping mechanisms to affect their mental well-being [50,51]. Humans improve their mood by alleviating the adverse effects of stress in a natural environment. The natural environment is closely related to people’s perception of a therapeutic environment, as well as an improvement in mood [52]. This study aims to examine the relationship between stress and mental well-being, which is a mediating variable in people’s perception of a recovery environment.**Hypothesis 5.** Forest recreation motivation will mediate the effect of stress on mental well-being. Motivation is the driving force to achieve or maintain a goal [53], and self-determination motivation leads to positive psychological function improvement. Therefore, people with self-determination motivation encourage their own performance, persistence, and well-being improvement [54]. Motives for engaging in forest recreation include having high expectations for health promotion, and its effects on mental and social health. The natural environment can contribute to mental health and well-being through participation in activities and obtaining benefits [26,27]. For forest visits, natural beauty is the essential motive for visiting [34], and fulfillment of this motive affects mental well-being. This study aims to verify the motivation for forest recreation, a factor in the relationship between stress and well-being.

### 2.4. Analysis Method

To test the hypothesis of this study, SPSS 26.0 and AMOS 18.0 were used, and the analysis method and procedure were as follows. A frequency analysis was performed to determine the general characteristics of forest users, and confirmatory factor analysis, reliability analysis, and correlation analysis were performed to examine the validity and reliability of the measurement variables that were used for the hypothesis testing. Mean, standard deviation, skewness, and kurtosis were calculated to check the general tendency and normality of the variables of this study, including stress, perceived restorativeness, forest recreation motivation, and mental well-being. The variable skewness ranged from −0.29~0.20, and the kurtosis range was from −0.40~0.50, which satisfied skewness < 3 and kurtosis < 10, indicating that all variables were normally distributed [55]. To test the hypothesis, the path coefficient of the research model was verified by applying the structural equation, and bootstrapping was performed to verify the mediating effect. A multi-group analysis was performed to confirm whether there was a difference in the path coefficient of the research model according to gender.

## 3. Results

### 3.1. Demographic and Sociological Characteristics of Study Subjects

In this study, a total of 1196 samples were collected from Mt. Gwanak (*n* = 385); Mt. Bukhan (*n* = 493); and Mt. Yumyeong (*n* = 318), from a total of 613 males (51.3%) and 583 females (48.7%). The largest age groups of forest users were 202 men in their 60s (33.0% of men) and 215 women in their 50s (36.9% of women). The frequency of forest visits was high, with 287 men (46.8%) and 237 women (40.7%) visiting once or twice a week. As companions for forest visits, the family was found to be high with 218 males (35.6%) and 263 females (45.1%). Males (28.2%) visited alone more frequently than females (19.0%), and it was found that women (31.6%) visited the forest with friends more often than men (29.2%). The most common length of time spent in the forest was ‘1–3 h’, with 281 males (45.8%) and 274 females (47.0%). When visiting the forest, as the main activity, climbing/walking was the highest with 523 men (60.8%) and 520 women (59.9%). As for the benefits of the forest, clean air was found to be the most important, with 402 males (32.9%) and 401 females (31.3%). Other general characteristics are presented in Table 1.

### 3.2. Descriptive Statistics and Correlation Analysis between Major Variables

As a result of Pearson’s correlation analysis between variables, stress had the highest negative correlation with mental well-being (*r* = −0.46, *p* < 0.001), and forest recreation motivation had a positive correlation with mental well-being (*r* = 0.31, *p* < 0.001). The Pearson correlation coefficient (r) ranged from −1 to 1 and was confirmed to be significant in this study as −0.46 < *r* < 0.31, *p* < 0.001. In conclusion, because of Pearson’s correlation analysis, stress was negatively correlated with perceived restorativeness, forest recreation motivation, and mental well-being. Perceived restorativeness was found to correlate with forest recreation motivation and mental well-being positively. Forest recreation motivation was positively associated with mental well-being (Table 2).

### 3.3. Differences in Stress Levels according to Demographic Characteristics

A chi-squared test was performed to verify the difference in stress levels according to the demographic characteristics of the subjects of this study. As a result, it was found that 7.9% belonged to the ‘healthy group’; 82.5% to the ‘potential stress group’; and 9.5% to the ‘high-risk group’ (Table 3). The difference in stress level according to the age (χ² = 20.50, *p* < 0.05) of forest users and the time spent visiting the forest (χ² = 16.82, *p* < 0.05) was found to be significant. Specifically, there was a significant difference in the high-risk group in their 40s. There was a significant difference in the ‘high-risk group’ when the length of stay in the forest was less than 30 min or between 30 min to 1 h, and a significant difference was found in the health group when the duration of stay in the forest was less than 5 h. On the other hand, there were differences in stress levels according to gender, frequency of forest visits, and companions who visited the forest, but they were not significant.

### 3.4. Confirmatory Factor Analysis

First, to verify the measurement model through confirmatory factor analysis, item parceling of the stress variables (18 items) consisting of a single factor was performed. Confirmatory factor analysis is an analysis estimate for individual items. As the number of items increases, the number of unknowns increases, and the estimation error increases. In a previous study [56], item grouping was performed, therefore, in this study, 18 items were created, and 3 item bundles of 6 items were created.

Next, the fit index of the confirmatory factor analysis model was confirmed. For the fit of the model, the absolute fit indices χ² and RMSEA [57], and the incremental fit indices IFI [58], TLI [59], and CFI [60] were used, and the acceptance criteria for the fitness index are presented, in Table 4. As shown in Table 4, the measurement model fit of this study was IFI = 0.943; TLI = 0.929; CFI = 0.943; and RMSEA = 0.072, all of which met the minimum standards, indicating that the model was statistically suitable.

As a result of the confirmatory factor analysis, the measurement model’s standardized regression coefficient (factor load) was 0.85 to 0.87 for stress; 0.40 to 0.84 for recovery environment perception; 0.57 to 0.86 for forest recreation; and 0.80 to 0.90 for mental well-being. Since the standardized regression coefficients of all the measured variables were above the minimum standard of 0.4 that was suggested by Kim [61], it can be seen that the measurement model composed of 15 measured variables and four latent variables is valid (Table 5).

### 3.5. Hypothesis Verification through Research Model Verification

As a result of the fit index of the model to the research model, as shown in Table 6, IFI = 0.935; TLI = 0.919; CFI = 0.935; and RMSEA = 0.076, indicating that the model was statistically suitable.

According to the parameter estimates of the research model (Figure 3), stress is related to the perceived restorativeness (*β* = −0.46, *p* < 0.001); forest recreation motivation (*β* = −0.26, *p* < 0.001); and mental well-being (*β* = −0.44, *p* < 0.001), and each was found to be significantly explained by negative. In other words, it was confirmed that the higher the stress, the lower the perceived restorativeness, forest recreation motivation, and mental well-being. Therefore, it was found that Hypothesis 1-1, which stated that stress would have a negative effect on the perceived restorativeness; Hypothesis 1-2, which indicated that stress would have a negative impact on the forest recreation motivation; and Hypothesis 1-3, which stated that stress would have a negative effect on mental well-being were accepted. The perceived restorativeness was positively significant for mental well-being (*β* = 0.16, *p* < 0.001), and it was confirmed that the higher the perceived restorativeness, the higher the mental well-being. In addition, it was found that Hypothesis 2, which stated that perceived restorativeness would have a positive effect on mental well-being was correct. The forest recreation motivation was positively significant for mental well-being (*β* = 0.22, *p* < 0.001), and it was confirmed that the higher the forest recreation motivation, the higher the mental well-being. Therefore, Hypothesis 3, which stated that forest recreation motivation positively affects mental well-being was correct (Table 7).

Next, the indirect effect was estimated by performing bootstrapping. Bootstrapping randomly restored and extracted 2000 virtual samples from the original data. The 90% confidence intervals (CI) estimated using bootstrapping were significant at the significance level of 0.1 when 0 was not included. As shown in Table 8, the indirect pathway (*B* = −0.07, CI: −0.12 to −0.04) in which stress affects mental well-being through the perceived restorativeness, or the indirect pathway (*B* = −0.06, CI: −0.11 to −0.06) in which stress affects mental well-being through forest recreation motivation was found to be significant. Since stress directly affects mental well-being, perceived restorativeness and forest recreation motivation partially mediate the relationship between stress and mental well-being. Therefore, Hypothesis 4, which sets the relationship between stress and mental well-being as the perceived restorativeness, and Hypothesis 5, which states that the relationship between stress and mental well-being will mediate the relationship between stress and mental well-being, are correct. The effect of stress on mental well-being was calculated to investigate the effects of perceived restorativeness and forest recreation motivation in more detail. The direct effect is the magnitude of the direct effect of one latent factor on another potential factor without going through any latent factor; the indirect effect is the value multiplied by the intermediate path, and the total effect is the value of the direct effect plus the indirect effect. Because of the analysis, as for the indirect effect of stress on mental well-being, the perceived restorativeness (−0.07) had a negatively higher indirect effect than the forest recreation motivation (−0.06), and the total effect was high. This indicates that the indirect effect of perceived restorativeness on the effect of stress on mental well-being was increased.

### 3.6. Multi-Group Analysis

A multi-group analysis was conducted to verify whether the path analysis model of forest users’ stress on mental well-being was suitable for other samples. In a multi-group analysis, the research model must satisfy the assumption of morphological identity and measurement identity in a sequence. The morphological identity was verified by classifying the data by gender and confirming whether the factor structure of the male (51.3%) and female (48.7%) groups was the same. Morphological identity means that if the model fit index is good in all groups then the shape identity assumption is satisfied. As the fitness index of each group is presented in Table 9, the male group had IFI = 0.953; TLI = 0.941; CFI = 0.952; and RMSEA = 0.068, and the female group had IFI = 0.931; TLI = 0.914; CFI = 0.931; and RMSEA = 0.077, confirming that the fitness index met the minimum standard and indicating that the shape identity assumption was satisfied.

Next, the measurement identity was verified. Measurement identity shows whether the same tool was used for measurement. It is confirmed by demonstrating the χ² difference between the constrained model that applied the equalization constraint to the factor load of the measured variable between the gender groups, and the free model that did not use the equalization constraint. If there is no significant difference in the fit between the two models, it is judged that the measurement equality assumption is satisfied. As the χ² difference test results are presented in Table 10, the difference in the χ² fit between the models is 9.67, which is not significant in the difference in degrees of freedom 11, so the measurement equality is satisfied.

Figure 4 and Table 11 show the results of the χ² test by applying the equalization constraint to each path coefficient, in order to verify whether there was a difference in the path coefficients according to gender since the assumptions of morphological identity and measurement identity were satisfied. As a result of the analysis, it was found that there was a significant difference in the effect of the perceived restorativeness on mental well-being (χ² = 6.95, *p* < 0.01) and the impact of forest recreation motivation on mental well-being (χ^2^ = 6.72, *p* < 0.05). Specifically, it was confirmed that the influence of perceived restorativeness and forest recreation motivation on mental well-being was more significant in the male group than in the female group. On the other hand, it was found that the difference in the effect of stress on the perceived restorativeness, the forest recreation motivation, and the mental well-being according to gender was not significant.

## 4. Discussion

This study revealed that stress was negatively correlated with perceived restorativeness, forest recreation motivation, and mental well-being. In addition, it was confirmed that the perceived restorativeness and the forest recreation motivation partially mediated the relationship between stress and mental well-being, and the direct and indirect effects of stress on mental well-being were confirmed. Moreover, it was confirmed that there was a difference in influence according to gender.

### 4.1. The Stress of Forest Visitors

The stress level of forest users, the subjects of this study, was found to be 8% in the health group, 82.5% in the potential stress group, and 9.5% in the high-risk group. Looking at previous studies, in the study of office workers by Chang [30], the high-risk group for stress was 22%; in the online survey for office workers by Kim [62], the high-risk group for stress was 25.5%; and in a study of Seoul citizens by the Seoul Institute [9], the highly stressed group during the COVID-19 period was found to be 14.9%. In this study, the proportion of the high-risk group for stress was lower than in previous studies. These results came from the fact that 50.2% of forest users in this study visited the forest at least once a week. It can be seen that exposure to nature, that is, the frequency of visits, contributed to personal psychological benefits and well-being [63].

There was also a significant difference in the length of stay in the forest by the stress level. In particular, the ‘health group’ showed the highest rate at more than 5 h, and the ‘high-risk group’ showed the highest rate at less than 30 min. Through this, it was found that there was a difference in the amount of time spent in the forest according to the stress level. In addition, exposure to the natural environment directly affects the stress hormone cortisol, and the natural environment reduces stress and enhances immunity [17,52,64,65]. Through many previous studies, it is possible to infer that stress is reduced in the natural environment. Since the subject of this study was a person descending from the forest and engaged in forest activities, their stress was reduced. In this study, 8.2% of men were in the high-risk group, and 11% of women were in the high-risk group, confirming that women had higher stress. Previous studies have shown that women are more stressed than men regarding mental health [66]. In addition, among forest users during COVID-19, the female group was more stressed than the male group. For example, women reported a higher prevalence of anxiety, depression, and stress due to the spread of COVID-19 [67], and female college students experienced more negative emotions [68].

### 4.2. Hypothesis Test

The success of Hypothesis 1-1 showed that stress negatively correlated with perceived restorativeness. This means that users who visited the forest experienced a reduction in stress. Stress resilience increases in people who visit a forest multiple times. The phytoncide that is released by trees reduces stress hormones and increases immunity by activating NK cells [13,15,69]. Therefore, in this study, it can be concluded that the stress of forest users and the perceived restorativeness are negatively correlated; in other words, stress is reduced and perceived restorativeness is increased.

The success of Hypothesis 1-2 showed that stress was negatively correlated with forest recreation motivation. Previous studies reported that recreational activities such as canoeing, backpacking, mountain climbing, caving, and hunting to relieve stress are highly motivating and bring satisfaction to participants [70]. Leisure acts as a buffer between stressors. Leisure activities that are not emotionally, cognitively, or physically burdensome and are enjoyed alone or with others can remove people from stressors or give them a temporary break. Light leisure activities in stressful situations provide health and wellness benefits. Leisure activities satisfy the motivation that is needed to cope with stressful situations and positively impact a person’s life [71,72]. Therefore, in this study, since stress is negatively correlated with forest recreation motivation, it can be concluded that forest users’ stress is reduced, and forest recreation motivation is increased.

The success of Hypothesis 1-3 showed that stress was negatively correlated with mental well-being. This was consistent with the research results showing that stress is negatively associated with well-being and psychological well-being. For example, it has been reported that stress in old age threatens the happiness of the elderly and is negatively correlated with well-being [45,73]. It was found that the more exposure to the natural environment, the higher the psychological well-being; while the higher the stress, the lower the psychological well-being [74]. Therefore, since there is a negative correlation between stress and mental well-being in this study, it can be concluded that the stress of forest users is reduced, and mental well-being is increased.

The success of Hypothesis 2 showed that perceived restorativeness had a positive correlation with mental well-being. Previous studies reported that exposure to nature has a positive relationship with emotional and psychological well-being (PWB) and that there are differences in the effect on the level of exposure to the recovery environment. [23,25]. Therefore, since there is a positive correlation between perceived restorativeness and mental well-being in this study, it can be concluded that the perceived restorativeness of forest users and mental well-being is increased.

Hypothesis 3: it was found that forest recreation motivation positively correlated with mental well-being. Visitors’ participation in recreation contributed to their mental health and well-being, and motivation came in the form of use. This leads to user satisfaction with the user experience. Motivational satisfaction is related to well-being, and motivation and well-being are positively correlated [27,34,53,71].

Hypothesis 4: Perceived restorativeness mediates the relationship between stress and mental well-being. Stress directly affects mental well-being, and the indirect effect on mental well-being through perceived restorativeness is significant. Since the impact of the perceived restorativeness on mental well-being was smaller than the direct effect, it was found that perceived restorativeness partially mediated the relationship between stress and mental well-being.

Hypothesis 5 showed that stress directly affected mental well-being, and the indirect pathway that affected mental well-being through forest recreation motivation was significant. Moreover, because the effect of forest recreation motivation on mental well-being was smaller than the direct effect, it was found that the forest recreation motivation partially mediated the relationship between stress and mental well-being.

Based on the above evidence, it can be concluded that when forest users reduce their level of stress in the forest environment, their perceived restorativeness, forest recreation motivation, and mental well-being all increase.

This structural equation derived the effect decomposition table from the research model.
Direct effect of stress → mental well-being −0.44, 
Stress → Perception of the recovery environment → 
The indirect effect on mental well-being −0.07, 
Total Effect −0.51

By deducing that forest users’ stress is reduced in the natural environment, if the stress decreases by 1, there is a direct effect that increases by 0.44 on mental well-being. It can be concluded that the indirect impact of the perceived restorativeness increases by 0.07 and increases by 0.51 in total. Additionally, suppose the stress of forest users decreases by 1. In that case, there is a direct effect that increases by 0.44 on mental well-being, and it can be concluded that the indirect effect of the motive for forest recreation increases by 0.06 and increases by 0.50 in total. As a result, a research model can be derived that stress reduction in forest users increases their perceived restorativeness, forest recreation motivation, and mental well-being.

### 4.3. Multi-Group Analysis

We conducted a multi-group analysis to verify whether there is a difference in path coefficients according to gender through a research model suggesting that forest users’ stress reduction increases their perceived restorativeness, forest recreation motivation, and mental well-being. As a result of this study, there were significant differences in perceived restorativeness on mental well-being and the effect of forest recreation motivation on mental well-being according to gender. Specifically, it was confirmed that the influence of forest recreation motivation and perceived restorativeness on mental well-being was more significant in the male group than in the female group. In addition, men showed a higher direct effect of stress on perceived restorativeness, and women showed a higher direct impact of stress on mental well-being. In the study of subjects who experienced the forest, men had a heightened perception of the forest recovery environment [75]. There was a positive relationship between men who were engaged in physical activity and their psychological well-being [76]. A lifestyle that supports well-being also differs according to gender [77], based on these previous studies; thus, the following conclusions can be drawn. The effect of stress on mental well-being did not differ by gender. Still, the effect of perceived restorativeness and forest recreation motivation on mental well-being was higher in men than in women.

### 4.4. Limitations

First, this study is meaningful because it was a face-to-face study conducted on forest users when studies showed that public stress was high due to anxiety about COVID-19 transmission and the absence of a vaccine. However, there were limitations, as it was carried out while practicing social distancing in a limited environment. Second, there was a limit to the psychological research that was conducted with the survey during the COVID-19 period. Therefore, in future studies, pre-and post-test verification studies and physiological effects studies should be conducted. Third, differences were verified through a multi-group analysis of gender in the research model for stress, perceived restorativeness, forest recreation motivation, and mental well-being. However, it is expected that new studies will be conducted through analysis between various groups based on the research model of future research. In addition, since stress reduction in forest users can increase their perceived restorativeness, forest recreation motivation, and mental well-being, various studies on the forest environment and forest healing for stress reduction should be conducted. In conclusion, this study has significance in developing a research model suggesting that forest users’ stress reduction leads to improvement in their perceived restorativeness, forest recreation motivation, and mental well-being. It is expected to serve as primary data for finding ways to promote forest healing in the future.

### 4.5. Implications

People avoid contact with others and travel short distances with their families due to COVID-19. It has become important for people to manage stress and lead a healthy life. This study found that perceived restorativeness and forest recreation motivation were mediators in reducing the stress of forest users and increasing their mental well-being. Moreover, there was a difference in the influence of men’s perceived restorativeness and forest recreation motivation on mental well-being, compared to women. To increase the perceived restorativeness and the forest recreation motivation among forest users, it is necessary to manage forest recreation in a way that is suitable for all users and various types of use. In addition, to increase perceived restorativeness, a natural environment that considers the factors of Being Away, Fascination, Coherence, and Compatibility is required.

## 5. Conclusions

This study is significant because it confirmed that the forest is a recovery environment that reduces stress and increases the perceived restorativeness, forest recreation motivation, and mental well-being of those citizens who visited the forest during the COVID-19 pandemic. Therefore, forest recreation management that can increase forest perceived restorativeness and forest recreation motivation is an important consideration for health promotion in the natural environment.

## Figures and Tables

**Figure 1 ijerph-19-06675-f001:**
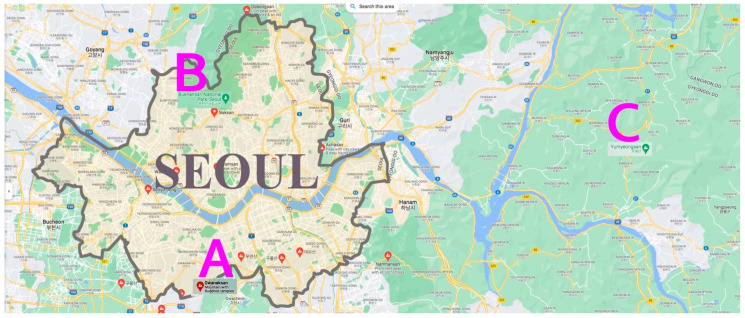
Research sites are located in the suburbs of Seoul. (**A**) Mt. Gwanak Urban Forest; (**B**) Mt. Bukhan National Park; (**C**) Mt. Yumyeong Recreation Forest (https://www.google.co.kr/maps/search/Mountain?hl=en) (accessed on 27 May 2022).

**Figure 2 ijerph-19-06675-f002:**
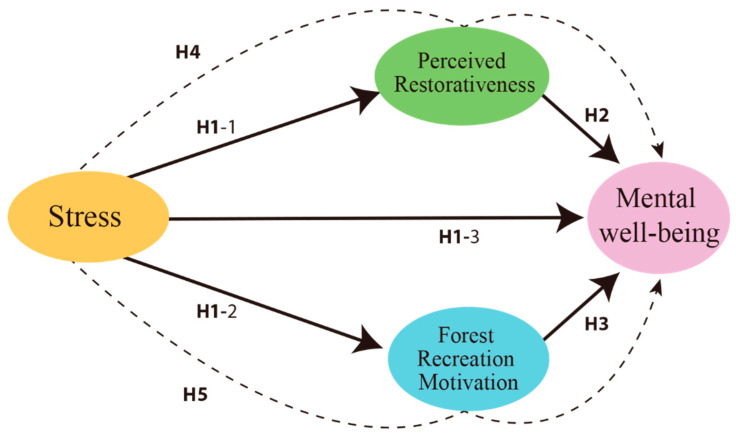
Research model.

**Figure 3 ijerph-19-06675-f003:**
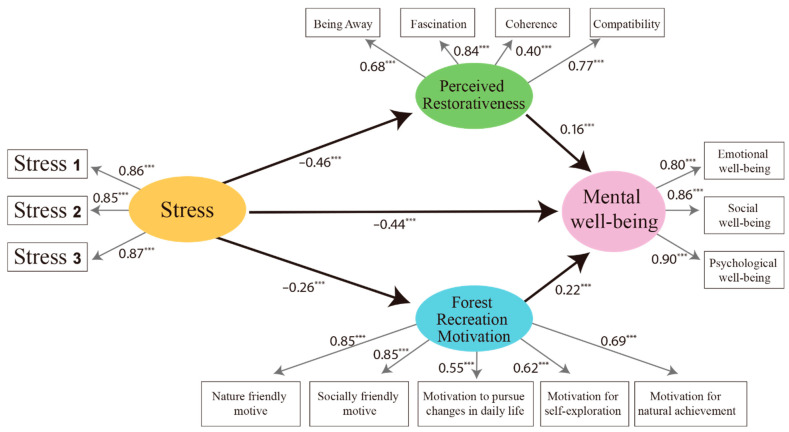
Research model path coefficient (standardized regression coefficient). *** *p* < 0.001.

**Figure 4 ijerph-19-06675-f004:**
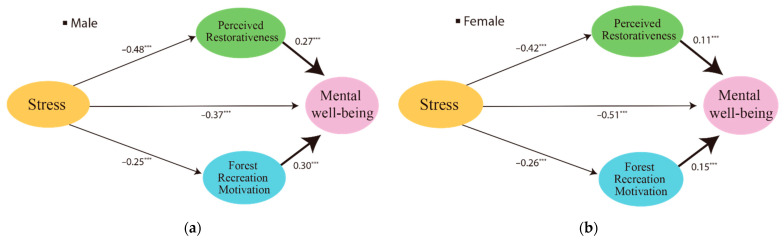
The difference in path coefficient according to male (**a**) and female (**b**). *** *p* < 0.001.

**Table 1 ijerph-19-06675-t001:** Descriptive characteristics of the study subjects.

Variable		Male *n* = 613 (%)	Female *n* = 583 (%)	Total *n* = 1196 (%)
Mountain	Mt. Gwanak	176 (28.7)	209 (35.8)	385 (32.2)
	Mt. Bukhan	281 (45.8)	212 (36.4)	493 (41.2)
	Mt. Yumyeong	156 (25.4)	162 (27.8)	318 (26.6)
Age	Less than 30	48 (7.8)	38 (6.5)	86 (7.2)
	30–39	53 (8.6)	48 (8.2)	101 (8.4)
	40–49	93 (15.2)	99 (17.0)	192 (16.1)
	50–59	165 (26.9)	215 (36.9)	380 (31.8)
	60–69	202 (33.0)	152 (26.1)	354 (29.6)
	More than 70	52 (8.5)	31 (5.3)	83 (6.9)
Forest visit frequency	Almost every day	38 (6.2)	38 (6.5)	76 (6.4)
	1–2 times/week	287 (46.8)	237 (40.7)	524 (43.8)
	1–2 times/month	178 (29.0)	161 (27.6)	339 (28.3)
	1–2 times/6 months	57 (9.3)	77 (13.2)	134 (11.2)
	1–2 times/year	38 (6.2)	47 (8.1)	85 (7.1)
	Rarely	15 (2.4)	23 (3.9)	38 (3.2)
People coming together into the forest	Alone	173 (28.2)	111 (19.0)	284 (23.7)
	Friends	179 (29.2)	184 (31.6)	363 (30.4)
	Colleagues	29 (4.7)	21 (3.6)	50 (4.2)
	Family	218 (35.6)	263 (45.1)	481 (40.2)
	Other	14 (2.3)	4 (0.7)	18 (1.5)
Time staying in the forest	Less than 30 min	20 (3.3)	22 (3.8)	42 (3.5)
	0.5–1 h	91 (14.8)	115 (19.7)	206 (17.2)
	1–3 h	281 (45.8)	274 (47.0)	555 (46.4)
	3–5 h	82 (29.7)	143 (24.5)	325 (27.2)
	More than 5 h	39 (6.4)	29 (5.0)	68 (5.7)
Activities in the forest (duplicate response)	Climbing/walking	523 (60.8)	520 (59.9)	1043 (60.4)
	Visiting cultural properties	106 (12.3)	100 (11.5)	206 (11.9)
	Cultural property viewing	13 (1.5)	12 (1.4)	25 (1.4)
	Relaxation/meditation	133 (15.5)	155 (17.9)	288 (16.7)
	Festival event	6 (0.7)	4 (0.5)	10 (0.6)
	Photography	33 (3.8)	42 (4.8)	75 (4.3)
	Collection of mineral water	11 (1.3)	6 (0.7)	17 (1.0)
	Use of sports facilities	20 (2.3)	23 (2.6)	43 (2.5)
	Environment commentary	6 (0.7)	1 (0.1)	7 (0.4)
	Other	9 (1.0)	5 (0.6)	14 (0.8)
Advantages of visiting the forest (duplicate response)	Fresh air	402 (32.9)	401 (31.3)	803 (32.1)
	Nature sounds and tranquility	227 (18.6)	245 (19.1)	472 (18.9)
	Beautiful scenery	166 (13.6)	200 (15.6)	366 (14.6)
	Scent of nature	180 (14.7)	196 (15.3)	376 (15.0)
	Refreshing from activity	241 (19.7)	231 (18.0)	472 (18.9)
	Other	6 (0.5)	7 (0.5)	13 (0.5)

**Table 2 ijerph-19-06675-t002:** Correlation analysis between key variables.

		Stress	Perceived Restorativeness	Forest Recreation Motivation	Mental Well-Being
Stress		1			
Perceived restorativeness		−0.40 ***	1		
Forest recreation motivation		−0.22 ***	0.30 ***	1	
Mental well-being		−0.46 ***	0.29 ***	0.31 ***	1
	Male	2.01 ± 0.37	5.26 ± 0.78	3.43 ± 0.56	3.68 ± 0.99
M ± SD	Female	2.03 ± 0.38	5.36 ± 0.75	3.41 ± 0.59	3.77 ± 0.99
	Total	2.02 ± 0.37	5.31 ± 0.77	3.42 ± 0.58	3.73 ± 0.72
Skewness		−0.29	−0.23	−0.11	0.20
Kurtosis		0.50	−0.08	0.32	−0.40

Note. M (Mean), SD (Standard deviation). *** *p* < 0.001.

**Table 3 ijerph-19-06675-t003:** Differences in stress levels according to demographic characteristics.

Variable		Healthy Group *n* = 95 (7.9%)	Potential Stress Group *n* = 987 (82.5%)	High-Risk Group *n* = 114 (9.5%)	χ^2^, *p*
Sex	Male	50 (8.2)	513 (83.7)	50 (8.2)	2.77
	Female	45 (7.7)	474 (81.3)	64 (11.0)	0.250
Age	Less than 30	12 (14.0)	60 (69.8)	14 (16.3)	20.50
	30–39	8 (7.9)	86 (85.1)	7 (6.9)	0.025 *
	40–49	15 (7.8)	151 (78.6)	26 (13.5)	
	50–59	34 (8.9)	318 (83.7)	28 (7.4)	
	60–69	22 (6.2)	298 (84.2)	34 (9.6)	
	More than 70	4 (4.8)	74 (89.2)	5 (6.0)	
Forest visit frequency	Almost every day	8 (10.5)	63 (82.9)	5 (6.6)	9.09
	1–2 times/week	45 (8.6)	437 (83.4)	42 (8.0)	0.524
	1–2 times/month	20 (5.9)	285 (84.1)	34 (10.0)	
	1–2 times/6 months	11 (8.2)	107 (79.9)	16 (11.9)	
	1–2 times/year	8 (9.4)	66 (77.6)	11 (12.9)	
	Rarely	3 (7.9)	29 (76.3)	6 (15.8)	
Group	Alone	20 (7.0)	236 (83.1)	28 (9.9)	7.12
	Friends	20 (5.5)	308 (84.8)	35 (9.6)	0.523
	Colleagues	5 (10.0)	39 (78.0)	6 (12.0)	
	Family	48 (10.0)	390 (81.1)	43 (8.9)	
	Other	2 (11.1)	14 (77.8)	2 (11.1)	
Time in the forest	Less than 30 min	3 (7.1)	32 (76.2)	7 (16.7)	16.82
	0.5–1 h	10 (4.9)	171 (83.0)	25 (12.1)	0.032 *
	1–3 h	42 (7.6)	461 (83.1)	52 (9.4)	
	3–5 h	28 (8.6)	272 (83.7)	25 (7.7)	
	More than 5 h	12 (17.6)	51 (75.0)	5 (7.4)	

Note: 0–8 points were defined as a healthy group; 9–26 points, a potential stress group; and 27–54 points were defined as a high-risk group, * *p* < 0.05.

**Table 4 ijerph-19-06675-t004:** Goodness-of-fit index of the measurement model.

	χ^2^	*df*	IFI	TLI	CFI	RMSEA
Acceptance criteria for fitness index	-		0.90 Over	0.90 Over	0.90 Over	0.08 Less than
Goodness-of-fit coefficient	603.89 ***	84	0.943	0.929	0.943	0.072

Note. χ^2^ (Chi-Square test of model fit); *df* (degrees of freedom); IFI (incremental fit index); TLI (Tucker Lewis index); CFI (comparative fit index); RMSEA (root mean square error of approximation), *** *p* < 0.001.

**Table 5 ijerph-19-06675-t005:** Path coefficient of the measurement model.

Variable		B	SE	*β*	*t*
Stress	Stress 1	0.88	0.02	0.86	36.85 ***
	Stress 2	0.90	0.03	0.85	36.24 ***
	Stress 3	1.00		0.87	
Perceived restorativeness	Being away	2.01	0.16	0.68	12.27 ***
	Fascination	2.09	0.16	0.84	12.86 ***
	Coherence	1.00		0.40	
	Compatibility	2.26	0.18	0.77	12.71 ***
Forest recreation motivation	Nature-friendly motive	1.71	0.08	0.86	20.20 ***
	Socially friendly motive	2.01	0.10	0.84	20.01 ***
	Motivation to pursue changes in daily life	1.00		0.57	
	Motivation for self-exploration	1.42	0.08	0.62	16.79 ***
	Motivation for natural achievement	2.08	0.12	0.68	17.76 ***
Mental well-being	Emotional well-being	0.93	0.03	0.80	33.69 ***
	Social well-being	0.97	0.03	0.86	37.10 ***
	Psychological well-being	1.00		0.90	

Note. B = estimates; SE (standard error); *β* = standardized estimates; *t* = B/se; C.R (critical ration); *** *p* < 0.001.

**Table 6 ijerph-19-06675-t006:** The fit index of the research model.

	χ^2^	*df*	IFI	TLI	CFI	RMSEA
Acceptance criteria for fitness index	-		0.90 Over	0.90 Over	0.90 Over	0.08 Less than
Goodness-of-fit coefficient	679.44 ***	85	0.935	0.919	0.935	0.076

Note. χ^2^ (Chi-Square test of model fit); *df* (degrees of freedom); IFI (incremental fit index); TLI (Tucker Lewis index); CFI (comparative fit index); RMSEA (root mean square error of approximation); *** *p* < 0.001.

**Table 7 ijerph-19-06675-t007:** Estimates of the parameters of the research model.

Path	B	SE	*β*	*t*
H1-1. Stress → perceived restorativeness	−0.90	0.07	−0.46	−13.04 ***
H1-2. Stress → forest recreation motivation	−0.46	0.06	−0.26	−7.96 ***
H1-3. Stress → mental well-being	−1.03	0.08	−0.44	−12.22 ***
H2. Perceived restorativeness → mental well-being	0.17	0.04	0.16	4.80 ***
H3. Forest recreation motivation → mental well-being	0.30	0.04	0.22	7.26 ***

Note. B = estimates; SE (standard error); *β* = Standardized estimates; *t* = B/se, *** *p* < 0.001. Arrows (→) indicate the direction of influence.

**Table 8 ijerph-19-06675-t008:** Structural model effect decomposition table.

Path	Direct Effect	Indirect Effect	Total Effect	CI
H4. Stress → Perceived Restorativeness → Mental well-being	−0.44	−0.07	−0.51	−0.12~−0.04
H5. Stress → Forest Recreation Motivation → Mental well-being	−0.44	−0.06	−0.50	−0.11~−0.06

Note. CI (confidence interval). Arrows (→) indicate the direction of influence.

**Table 9 ijerph-19-06675-t009:** Model fit index by gender.

Group	χ^2^	*df*	IFI	TLI	CFI	RMSEA
Acceptance criteria for fitness index	-		0.90 Over	0.90 Over	0.90 Over	0.08 Less than
Male	318.88 ***	84	0.953	0.941	0.952	0.068
Female	376.38 ***	84	0.931	0.914	0.931	0.077

Note. χ^2^ (Chi-Square test of model fit); *df* (degrees of freedom); IFI (incremental fit index); TLI (Tucker Lewis index); CFI (comparative fit index); RMSEA (root mean square error of approximation); *** *p* < 0.001.

**Table 10 ijerph-19-06675-t010:** Goodness-of-fit index of free model and constrained model.

Model	χ^2^	*df*	IFI	TLI	CFI	RMSEA	Δ*df*	Δχ^2^
Acceptance criteria for fitness index	-		0.90 Over	0.90 Over	0.90 Over	0.08 Less than		
Freestyle model	695.27 ***	168	0.943	0.928	0.943	0.051	-	-
Constraint model	704.94 ***	179	0.943	0.933	0.943	0.050	11	9.67

Note. χ^2^ (Chi-Square test of model fit); *df* (degrees of freedom); IFI (incremental fit index); TLI (Tucker Lewis index); CFI (comparative fit index); RMSEA (root mean square error of approximation); *** *p* < 0.001.

**Table 11 ijerph-19-06675-t011:** Comparison of path coefficients between groups.

Path	β		Δχ^2^	*p*
	Male	Female		
Stress → Perceived restorativeness	−0.48 ***	−0.42 ***	2.30	0.129
Stress → Forest recreation motivation	−0.25 ***	−0.26 ***	0.14	0.708
Stress → Mental well-being	−0.37 ***	−0.51 ***	3.18	0.075
Perceived restorativeness → Mental well-being	0.27 ***	0.11 **	6.95 **	0.008
Forest recreation motivation → Mental well-being	0.30 ***	0.15 ***	6.72 *	0.010

Note. Arrows (→) indicate the direction of influence. * *p* < 0.05, ** *p* < 0.01, *** *p* < 0.001.

## Data Availability

The data presented in this study are available on request from the corresponding author. The data are not publicly available due to privacy.

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
