# Peer review of "Effect of Forest Users’ Stress on Perceived Restorativeness, Forest Recreation Motivation, and Mental Well-Being during COVID-19 Pandemic"

_ijerph, 2022, doi:10.3390/ijerph19116675_

Round 1

Reviewer 1 Report

Comments to Authors:

I really appreciate your work. But it is understood that the forests always have a great impact on human health.

Please concise the introduction part, it's too long with general talk. Summarize it in 5 to 6 paragraphs.

“This study aims to investigate” These words you have used many times. Please summarize it in the last paragraph of the introduction part.

Don’t need to write aims and research gaps in the Methodology section.

Don’t use repeated information, I mean the information which already has been explained, don’t repeat it.

You need to clarify your hypothesis after your research objectives in the last paragraph of the introduction part.

Please use the latest references and replace it with older ones.

Need to add climatic conditions of that area.

Which forest type exists there?

The authors carefully check the typos throughout the manuscript.

Authors should improve the language, if grammatical problems are solved, it will be of great help to reviewers and readers to easily understand your manuscript.

Author Response

Reviewer #1

Thank you for providing detailed comments. We marked the modified part in blue.

  1. We modified the introduction so that it’s concise (line 36-103)
  2. We deleted to aims and research gaps in the Methodology section.
  3. We deleted repeated information.
  4. We inserted our hypothesis after the research objectives in the last paragraph of the introduction part (line 95-103)
  5. We inserted the latest references.
  6. We added the climatic conditions of that area. (line 111-112)
  7. We added reference to our article for detailed information related to the location of this study (line 112, Figure 1)
  8. We carefully checked the typos throughout the manuscript.
  9. We modified the grammatical program.

Reviewer 2 Report

In this paper, authors study the relationship between stress, perceived restorativeness, forest recreation motivation, and the mental well-being of forest users who visited three mountains in the Seoul metropolitan and the Gyeonggi-do area. To do so, the authors propose five hypotheses on the relationship between these four components. While I appreciate the effort made by the authors, I think the article has serious flaws that need to be revised and improved.

GENERAL CONCERNS:

I will break down and discuss my major concerns:

  • In subsequent submitted manuscripts, please, I recommend adding continuous line numbers. This certainly facilitates peer review process.
  • In general, the manuscript is very disorganized, and this makes it difficult to read and understand. For instance, there are very long parts of the text (e.g. some parts of the introduction bear little relation to the rest of the manuscript), misplaced paragraphs, (e.g. methods in results, methods in results, aims in methods, etc.) and often very repetitive. I could quote several parts of the text but there are no line numbers to refer to (I add some cases in the paragraph "SPECIFIC COMMENTS").
  • The authors discuss four very broad concepts that can be interpreted in different ways. In addition, it should be noted that IJERPH is an interdisciplinary journal with readers of very different profiles. What the authors mean by each of the terms? In order to facilitate the interpretation of analyses and results, it is necessary to clearly define the concepts used.
  • Please be careful with references (or lack of them). Throughout the text (introduction and discussion) there are some sentences without any reference.
  • In my opinion the objectives paragraph is repetitive and unnecessarily long (e.g. methodological aspects such as structural equations or multi-group analysis can be eliminated). I also think that it does not reflect what the authors study. I recommend that you write a general objective sentence about the relationship between stress and perceived restorativeness, forest recreation motivation, and mental well-being and then state the 5 hypotheses (Fig. 1) in a summarized form.
  • I am also concerned about the reproducibility of the methods and finally the interpretation of the results. First, what kind of methodology did the authors follow to select the respondents (sampling methods) and to obtain the information (questionnaires, structured interviews, etc.). Is the selected sample representative of the total population? Second, there are many variables that appear for the first time as results (e.g. in Table 1), but how these answers were obtained? For instance, what does “time staying in the forest” mean? (total?, each visit?, etc.). To facilitate the interpretation of the results, I recommend adding a summary table with the name of each variable, the type (e.g. categorical) and the exact question that was asked during the surveys; this can be supplementary material.
  • Regarding methods, is not clearly justified because the authors conducted the analyses according to the gender (male vs female) of forest users, and not on the basis of other socio-demographic characteristics.
  • The first discussion paragraph should be a short summary of the main findings of the paper. Now, the first lines, it looks like a methods paragraph.
  • In the conclusions the authors should highlight the main conclusions in clear and simple language (do not simply reiterate your discussion with previous studies).

EXAMPLES:

Misplaced paragraphs (examples, but they are not the only ones): please move it

-“This study aims to investigate the relationship between stress, perceived restorativeness, forest recreation motivation, and mental well-being of forest users.” (page 4 to aims)

-”As a result of the survey, a total of 1,196 samples were collected from Mt. Gwanak (n=385), Mt. Bukhan (n=493), and Mt. Yumyeong (n=318), …” (page 5 to results)

-”Mean, standard deviation, skewness, and kurtosis were calculated to check the gen-eral tendency and normality of the variables of this study, including stress, perceived re-storativeness, forest recreation motivation, and mental well-being. ” (page 7 to methods)

-”Also, there was a significant difference in the length of stay in the forest by stress level (χ²= 16.82, p= .032). ” (page 14 to results)

- ”through structural equations. Also, through multi-group analysis according to gender, we want to find out if there is a statistically significant difference between gender…” (page 9 to methods)

Lack of references (examples): please add references

-“Forests occupy about 64% of Korea's land area. Since the 1980s, the government has been promoting forests as a place for leisure activities and outdoor recreation.” (page 2)

-“The forest environment indirectly improves immune functions by reducing stress hormone levels” (page 2)

Repetitive (examples): please remove repetitions through the ms

-“Therefore, since there is a negative correlation between stress and mental well-being in this study, it can be concluded that the stress of forest users is reduced and mental well-being is increased.” (page 15)

-“Since there is a positive correlation between the forest recreation motivation and mental well-being in this study, it can be concluded that the forest recreation motivation of forest users increases mental well-being.” (page 15)

SPECIFIC COMMENTS:

-In Hypothesis 4 and 5. What do the authors mean by mediating the effect of stress? Please explain it.

- Why these three places?, What characteristics do they have in common or different?, Can this influence the results?

- The results and especially the values of statistical analyses (e.g. χ²= 16.82, p= .032; β= -.45, p< .001, …); should be in results (not in discussion).

- Table 2. What represents the numerical value of stress, mental well-being, etc.? As I said before, each variable must be defined. Right now, it is impossible to understand these correlations.

- Table 3. Again, this is the first time the 3 stress levels are mentioned. This must be established and defined in methods.

- Figure 3. I recommend adding Male and Female to the figure itself (so that it is self-replicative without the need to read Figure captions).

Author Response

Reviewer #2

Thank you for providing detailed comments. We marked the modified part in green.

  1. We added continuous line numbers.
  2. We deleted repetitive content from the text and rearranged paragraphs so that readers can understand it.
  3. We added reference and line numbers to it.
  4. We moved or deleted the misplaced paragraphs.
    • “This study aims to investigate the relationship between stress, perceived restorativeness, forest recreation motivation, and mental well-being of forest users.” (page 4 to aims) (line 88-89)
    • ”As a result of the survey, a total of 1,196 samples were collected from Mt. Gwanak (n=385), Mt. Bukhan (n=493), and Mt. Yumyeong (n=318), …” (page 5 to results) (line 255-256)
    • ”Mean, standard deviation, skewness, and kurtosis were calculated to check the gen-eral tendency and normality of the variables of this study, including stress, perceived re-storativeness, forest recreation motivation, and mental well-being. ” (page 7 to methods) (line 243-248)
    • multi-group analysis according to gender, we want to find out if there is a statistically significant difference between gender…” (we deleted it)
  5. We added reference.
  6. We deleted repetitive content from the text
    • Therefore, since there is a negative correlation between stress and mental well-being in this study, it can be concluded that the stress of forest users is reduced and mental well-being is increased.”
    • “Since there is a positive correlation between the forest recreation motivation and mental well-being in this study, it can be concluded that the forest recreation motivation of forest users increases mental well-being.”
  7. We conducted three locations to survey various forest users during the Covid-19.
  8. We added results and especially the values of statistical analyses
  9. We added the contents of the measurement tool (ex, stress, mental well-being, etc) to the method (line 120-154)
  10. We revised Figure 3 (about males and females) (line 353).

Reviewer 3 Report

Effect of Forest Users’ Stress on Perceived Restorativeness, Forest Recreation Motivation, and Mental Well-being

This is an interesting article with important implications for the field. Nevertheless, I believe the following changes would improve the overall quality of the manuscript:

  1. Authors should rewrite the abstract. It is 340 words long and does not adhere to the journals template.
  2. Reference (2): Which cognitive resources? Please provide some examples.
  3. Please rephrase the paragraph “When we meet the gentle charm of the natural environment, we lose our gaze unwittingly…” Authors should be consistent with the use of the verbal form to be more objective and scientific (refrain from the use of the pronoun we).
  4. Include subtopics (1.1, 1.2, etc) in the introduction based on the information being mentioned (ex: benefits, motivations, etc).
  5. Please rewrite objectives to provide more clarity and refrain from repeating them at the end of the introduction and in 2.1.
  6. Authors must include a materials section, informing which instruments were used for each variable under study, including psychometric information, number of items, examples of items, etc.
  7. Table 3. Use * to inform level of significance.
  8. Authors should include an implications section. Please discuss the importance of your results in terms of public health.

Best wishes.

Author Response

Reviewer #3

Thank you for providing detailed comments. We marked the modified part in red.

  1. We rewrote the abstract and revised it. (line 13-30)
  2. We added some examples of reference (2) (line 39)
  3. We deleted the abstract expression and modified it.
  4. We revised it to objective and scientific verbal.
  5. We revised the introduction concisely, including the information (ex, benefits, motivations, etc). (line 13-30)
  6. We clarified the purpose of the study and modified it not to repeat at the end of the introduction and in 2.1.
  7. We added the contents of the measurement tool to the method. (line 120-154)
  8. We added Table 3. Use * to inform the level of significance.
  9. We added the application section. (line 563-572)

Reviewer 4 Report

The manuscript depicts an exhaustive study of the positive effects of forests in stress reduction and mental well-being, proposing a very interesting research model.

The plot of the paper is well designed, and the statistical approach are updated and straight.

Even methods section is very short and briefly described, the methodologies used and their details are clearly exposed in the results section and probably this change in the structure of traditional papers are justified.

I liked very much the section called “limitations” depicting the extent of the study and the future possibilities for continuing the research on the topic. But I miss in the title the main constraint, or specific conditions or limiting factor of the study, I mean it was performed during COVID pandemic.

I suggest include this relevant aspect in the title of the paper in any way, for instance, including after the current title “:COVID19 pandemic case study” or any other reference to that in the title.

Minor comments:

Table 1 “Time staying in the forest 1-3 h” correct 2%)45.8)

Table 1 and 2 (and in results section text) include “0” in all decimal numbers between 0 and 1 (see activities in the forest section of the table) for easy reading. In table 1 some are in 0.X way and others in the .X   ( i.e. -0.40 instead of -.40)

Define acronyms before use and include definitions in tables (4, 5,7) captions. (i.e. IFI TLI CFI RMSEA, B, SE, beta, t)

Author Response

Reviewer #4

Thank you for providing detailed comments. We marked the modified part in purple.

  1. We modified Table 1.”Time staying in the forest 1-3h” (Table 1. line 270)
  2. We modified Table 1 and Table 2 to include “0” in all decimal numbers between 0 and 1. (Table 1, Table 2)
  3. We added it to define acronyms before use and include definitions in table (4, 5, 7) captions. (i.e. IFI, TLI, CFI etc.) (Table 4~11)

Round 2

Reviewer 3 Report

Thank you for implementing all the requested changes. The article's quality has very much improved and is now fit for publication. Best wishes.